# The Multiple Promoting Effects of *Suaeda glauca* Root Exudates on the Growth of Alfalfa under NaCl Stress

**DOI:** 10.3390/plants13060752

**Published:** 2024-03-07

**Authors:** Linling Dong, Yi Hua, Zhiqiang Gao, Hanfu Wu, Yu Hou, Yingying Chu, Jinwei Zhang, Guowen Cui

**Affiliations:** College of Animal Science and Technology, Northeast Agricultural University, Harbin 150030, China; donglinll@163.com (L.D.); qq2549188890@163.com (Y.H.); gaozhiqiang202108@163.com (Z.G.); w995450205@sina.com (H.W.); 13359600161@163.com (Y.H.); 15336343174@163.com (Y.C.); zhangjw133@nenu.edu.cn (J.Z.)

**Keywords:** abiotic stress, physiological response, PLS-SEM, soil bacterial community, soil enzyme activity

## Abstract

Under abiotic stress, plant root exudates can improve plant growth performance. However, studies on the effect of root exudates on the stress resistance of another plant are insufficient. In this study, root exudates (REs) were extracted from *Suaeda glauca* to explore their effect on alfalfa seedlings under salt stress. The results showed that the plant height and fresh weight of alfalfa significantly increased by 47.72% and 53.39% after 7 days of RE treatment at a 0.4% NaCl concentration. Under 1.2% salt stress, REs reduced the Malondialdehyde content in alfalfa by 30.14% and increased the activity of its antioxidant enzymes (peroxidase and catalase) and the content of its osmotic regulators (soluble sugar and proline) by 60.68%, 52%, 45.67%, and 38.67%, respectively. Soil enzyme activity and the abundance of soil-beneficial bacteria were increased by REs. Spearman analysis showed that urease and neutral phosphatase were related to the richness of beneficial bacteria. Redundancy analysis confirmed that urease affected the composition of the soil bacterial community. The partial least squares structural equation model (PLS-SEM) revealed that REs had a direct positive effect on alfalfa growth under salt stress by regulating the plant’s injury and antioxidant systems, and the soil bacterial community had an indirect positive effect on alfalfa growth through soil enzyme activity.

## 1. Introduction

Root exudates (REs), a series of organic compounds and inorganic ions actively or passively released by plant roots during growth [1], play a crucial role in mediating material circulation and energy flow between the plant, soil environment, and rhizosphere microorganisms [2,3]. REs can chelate ions in the soil, such as iron, phosphorus, etc., to provide the required nutrients for plants [4]. For example, the carboxylates in RE can dissolve the absorbed phosphorus in plants [5]. Simultaneously, their combination with soil particles helps to form stable soil aggregates, maintain soil structure, and promote the growth of plant roots [6]. Under abiotic stress, REs can significantly improve plant performance, promoting growth and productivity [7,8,9]. Under low-phosphorus stress, *Stylosanthes guianensis* (Aubl.) Sw. secretes abundant organic acids and flavonoids from its roots, promoting the circulation and utilization of organic and insoluble phosphorus in the soil. This, in turn, enhances the nutrient absorption and physiological activity of the plant [10]. Similarly, when facing abiotic stresses, plants enhance their stress resistance by increasing the release of specific REs. Vicente et al. showed that, under salt and high-temperature stress conditions, the release of proline in the rhizosphere of citrus plants increased significantly. The increase in these REs is essential for regulating intracellular osmotic pressure and maintaining water balance, effectively protecting plant cells from damage [11]. In this way, the secretion of proline and other plant hormones not only directly participates in the physiological response of plants to abiotic stress, but also reflects that plants adapt to and survive in unfavorable environmental conditions through complex RE regulation mechanisms. In addition, stress-tolerant plants also increase the synthesis and accumulation of secondary metabolites such as polyphenols and flavonoids. These substances exhibit strong antioxidant activity, can scavenge free radicals, and reduce the damage caused by oxidative stress in plant cells [12]. For instance, the total phenolic content of *Capsicum annuum* L. increased under moderate saline irrigation [13], while the content of phenolic compounds in *Hypericum brasiliense* (Guttiferae) significantly increased under water stress [14]. The study of Dardanelli et al. showed that the concentration of flavonoids in the REs of soybean (*Glycine max* (L.) Merrill.) and kidney bean (*Phaseolus vulgaris* L.) increased significantly under salt stress, which further proved the coping strategies of plant roots to adversity [15,16]. Flavonoids not only play an anti-stress role in plants but also influence the soil environment through REs, promoting the interaction between plants and soil microorganisms and enhancing plant nutrient acquisition. Simultaneously, these compounds can regulate the biological and chemical properties of soil, including enzyme activity, nutrient cycling, and the microbial community, thereby positively impacting the overall quality and function of soil [17,18].

Providing carbon sources for soil microorganisms influences the activities and structures of beneficial microorganisms. These microorganisms, in symbiosis with mycorrhizal fungi, can enhance the absorption of nutrients by plants [19]. For example, maize can have a positive impact on nearby plants by recruiting beneficial rhizobia in the rhizosphere soil through its RE. These rhizobia promote *Zea mays* L. growth through nitrogen fixation, improving soil nutrients, regulating plant mechanisms, and providing defense to reduce the damage from pests and diseases [20]. Soybeans possess the ability to impede weed growth by disrupting the physiological processes of weeds. Additionally, soybeans release compounds that facilitate nutrient absorption and foster beneficial microbial interactions. These actions contribute to improved nutrient cycling, enhance the overall health of plants, and consequently alter the nutrient supply conditions within the soil [21]. At a low concentration of *Zea mays* L.’s RE, the transcription level of the chemoreceptor gene of Pseudomonas putida KT2440 will increase, while at a high concentration it decreases. Plant REs can adversely affect harmful bacteria [22]. Plant REs influence the entire microbial community by selectively inhibiting or promoting the growth of specific microorganisms [23]. For example, antibiotics or other antimicrobial compounds in REs inhibit the growth of pathogenic bacteria while promoting the proliferation of beneficial microorganisms [24]. This selective effect helps plants establish a microbial environment conducive to their own growth while inhibiting potential pathogens. Plant REs also influence the microbial diversity of rhizosphere soil. For example, plant REs can increase the microbial diversity of barren stony soil, making its microbial structure more similar to fertile soil [25]. Other studies have shown that REs can promote the emergence of complex and resilient microbial communities containing a large number of functional microorganisms [26].

Many studies on the effects of REs on plant stress resistance have focused on the plant itself and its own soil, and most studies have focused on the effects of an RE on soil microorganisms, which thus affect the growth of plants. However, REs are compositionally complex, encompassing not only carbon sources that are utilized as microbial nutrients but also plant hormones that directly influence plant growth, along with secondary metabolites exerting allelopathic effects on other plants [4,27,28]. Given the crucial role of REs in abiotic stress, we propose the hypothesis that REs from plants with strong stress resistance can promote the growth of another plant under stress by enhancing soil enzyme activity and improving the microbial community. If this hypothesis is scientifically verified, revealing the ways in which REs improve plant stress resistance will provide strategies for improving the stress resistance of key crops. This will not only enhance crop productivity but also offer a more sustainable and environmentally friendly approach to addressing the environmental problems caused by global climate change.

In order to verify the above hypothesis, we explored the effects of *Suaeda glauca* REs on the growth of alfalfa under salt stress. Approximately 935 million hectares of global agricultural land are estimated to be threatened by salinization [29,30]. Additionally, the area of salinized soil is expanding at a rate of 1 to 2 million hectares per year. With the frequent occurrence of climate change, especially extreme drought and flood events, the rate of soil salinization is expected to increase further in the coming decades, becoming one of the most concerning soil and environmental issues globally [31,32,33]. The global soil salinization problem is driven by various natural factors and human activities, including changes in soil and landscape characteristics, frequent waterlogging and coastal floods, high evaporation, insufficient rainfall, and unsustainable agricultural practices with inappropriate irrigation methods [34,35,36]. In recent years, with the continuous growth of the population and the scarcity of cultivated land resources, increasing amounts of wasteland have been converted into cultivated land. Consequently, the impact of soil salinization caused by irrigated agricultural land is becoming more intense [37,38]. By 2050, the world population is expected to exceed 9 billion, necessitating a 57% increase in food production. The decline in agricultural productivity due to soil salinization poses a significant challenge to our agricultural capacity to support a growing population [39]. *Suaeda glauca* (Bunge) Bunge is an annual herbaceous salt-accumulating plant belonging to the genus Suaeda. It has strong salt tolerance and can grow in soils with a salt content higher than 0.48%. It is regarded as a model plant for studying the salt tolerance of plants [40,41]. It does not only survive in high-salt environments, but also plays a role in maintaining and stabilizing a salinized-soil ecological environment [42]. Alfalfa (*Medicago sativa* L.) is an excellent perennial forage legume, which is widely planted all over the world due to its high nutritional value and wide adaptability [43,44]. Many studies have confirmed that alfalfa has a certain salt tolerance. When the NaCl concentration is less than 40 mmol/L, its growth is not inhibited, but when the NaCl concentration is higher than 200 mmol/L, the growth of alfalfa is inhibited [45,46]. Therefore, the growth and productivity of alfalfa in high-salinity environments are severely limited [44]. This study aims to investigate whether the REs of *Suaeda glauca* can promote the growth of alfalfa; if so, what is the mechanism of this promotion?

## 2. Results

### 2.1. Changes in the Growth Parameters of Alfalfa

In order to verify the effects of the *Suaeda glauca* RE treatment, the plant height and shoot fresh weight of alfalfa under salt stress were measured (Figure 1). After 1 day of the RE treatment, the plant height and fresh weight of alfalfa were significantly increased (*p* < 0.05). At 7 day, under the 0% NaCl (CK) and 0.4% NaCl concentrations, the plant height of alfalfa treated with REs increased by 76.57% and 47.70%, respectively, and its fresh weight increased by 41.38% and 50.00%, respectively. However, under the 1.2% NaCl concentration, only the fresh weight increased significantly (*p* < 0.05), and there was no significant change in plant height. At 14 day, the plant height of alfalfa was significantly increased by 70.05% and 39.06% (*p* < 0.05) under the CK and 0.4% NaCl concentration, respectively, but there was no significant change in plant height under the 1.2% NaCl concentration. However, the effects of the RE treatment on fresh weight were not significant at either the 0.4% or 1.2% NaCl concentration.

### 2.2. Physiological Changes in Alfalfa

The RE treatment reduced the Malondialdehyde (MDA) content of alfalfa leaves (Figure 2a). Especially under 14 day of high salt (1.2% NaCl) stress, the MDA content of the RE treatment group decreased by 30.14% compared with that of the no treatment group.

In this study, the changes in the antioxidant enzyme activity in alfalfa were determined (Figure 2b–d). The RE treatment increased the antioxidant enzyme activity of alfalfa leaves. Especially at 7 day, the RE treatment significantly increased the antioxidant enzyme activity of alfalfa leaves at the 0.4% and 1.2% NaCl concentrations (*p* < 0.05). Specifically, peroxidase (POD) activity increased by 52.51% and 31.21%, superoxide dismutase (SOD) activity increased by 67.42% and 57.08%, and catalase (CAT) activity increased by 45.40% and 47.21%, respectively. However, at a 1.2% NaCl concentration for 14 day, the RE treatment significantly increased POD and CAT enzyme activities (*p* < 0.05), but did not significantly increase SOD activity.

In terms of soluble sugar and proline content, the RE treatment and alfalfa leaf antioxidant enzyme activity change trend are consistent, and can almost improve their contents (Figure 2e,f). At 7 day and 14 day, under 1.2% NaCl stress, the soluble sugar content of the RE treatment was increased by 54.88% and 45.67%, respectively, and the proline content was increased by 49.31% and 38.67%, respectively.

### 2.3. Changes in the Enzyme Activities in the Rhizosphere Soil of Alfalfa

The results showed that the activities of soil urease (S-UE), soil sucrase (S-SC), and soil catalase (S-CAT) at the 1.2% NaCl concentration were significantly lower than those of the CK at 1 day (Figure 3). At 14 day, with the increase in salt concentration, the activities of S-UE, S-SC, and soil neutral phosphatase (S-NP) under the 1.2% NaCl concentration increased significantly compared with the CK (*p* < 0.05). The RE treatment had an effect on a variety of enzyme activities. At 1 day, the REs decreased the S-UE activity of the CK, but increased the S-UE activity under 1.2% NaCl. At the same time, the S-SC activity of the CK also increased significantly. However, at 14 day, although the REs changed the enzyme activity of S-UE and S-SC, this difference was not significant. In particular, REs could significantly increase the S-CAT activity of the CK at 14 day (*p* < 0.05), while, under other conditions, the S-CAT activity also increased, although not significantly, compared to untreated plants. Compared with other enzymes, REs have little effect on S-NP.

### 2.4. Changes in the Rhizosphere Bacterial Community in Alfalfa

#### 2.4.1. Bacterial Community Diversity

Species diversity (Shannon) and bacterial species richness (Observed_otus and Chao1) were compared. The results showed that no matter whether under normal growth conditions or under high salt stress conditions, the application of *Suaeda glauca* REs and no application of *Suaeda glauca* REs had no significant effects on the bacterial diversity index of rhizosphere soil (Figure 4a).

#### 2.4.2. Bacterial Community Composition and Structure

The phylum and genus of the top 30 most abundant bacteria were selected to show the taxonomic composition of the bacterial community (Figure 4b,c). A total of 83 phyla were detected in rhizosphere soil samples, and the top 6 were Proteobacteria (16.35~43.81%), Actinobacteriota (18.54~33.88%), Acidobacteriota (7.78~13.19%), Gemmatimonadota (5.62~13.14%), Planctomycetota (3.68~16.25%), and Chloroflexi (4.03~10.97%). Further data analysis showed that the Proteobacteria of the RE treatment increased by 0.72% compared to the CK, while the Proteobacteria of the 1.2% NaCl + RE treatment decreased by 3.22% compared to 1.2% NaCl (Figure 4b).

In order to further explore the effects of REs on the community structure of alfalfa rhizosphere bacteria, the community composition of the alfalfa rhizosphere bacteria was analyzed at the genus level. The proportions of *Gaiellale* and *Pseudolabrys* in the RE treatment were 1.16% and 0.06% lower than the CK, respectively. On the contrary, the proportions of *Gaiellale* and *Pseudolabrys* in the 1.2% NaCl + RE treatment were 0.54% and 0.58% higher than the 1.2% NaCl, respectively (Figure 4c).

#### 2.4.3. Distribution of Bacterial Species Abundance

According to the species annotation and abundance information of all the samples at the genus level, we selected the 30 genera with the highest abundance and classified them into two levels according to the abundance of the species and in the samples (Figure 4d). From the diagram it can be seen that, in terms of differential changes in the dominant bacterial genera, the RE treatment group significantly reduced its relative expression abundance of *Bryobacter* and *Candidatus_Solibacter* in *Acidobacteriota* compared with the CK group, and increased its expression abundance of *Bradyrhizobium*, *Dokdonella*, and *SC-I-84_unclassified*. Compared with the 1.2% NaCl treatment group, the 1.2% NaCl + RE treatment group significantly reduced its relative expression abundance of *Hyphomicrobiaceae*, *Haliangium*, *Alphaproteobacteria*, *Vicinamibacterales*, *Sphingomonas*, and *Micropepsaceae*, and increased its expression abundance of *Gemmatimonas*, *Gemmata*, *WPS-2*, *KD4-96*, and *IMCC26256*.

### 2.5. Effects of REs on the Growth of Alfalfa Seedlings under Salt Stress

The validity of the PLS-SEM model was evaluated to effectively evaluate its reliability. Since the data in this study well meet the above conditions, the factor loads of the main indicators in Table A1 all meet the requirements of structural validity. A variance inflation factor (VIF) was used to detect the multicollinearity of 20 variables. The results showed that the VIF values of all dominant variables were between 1 and 5, indicating that there was no serious covariance between elements. The standardized root mean square residual (SRMR) values of the CK, RE, 1.2% NaCl, and 1.2% NaCl + RE treatments were 0.088, 0.097, 0.095, and 0.092, respectively, indicating that the model had a good degree of fitting. In summary, the above indicators have basically reached their ideal value, which proves the rationality and reliability of the evaluation model.

Figure 5 shows the partial least squares regression model of the relationship between the latent and dominant variables of the CK, RE, 1.2% NaCl, and 1.2% NaCl + RE treatments. The model explained 4.0%, 12.0%, 3.4%, and 7.3% of the variation in the CK, RE, 1.2% NaCl, and 1.2% NaCl + RE groups. This finding reflects the ability of REs to explain the changes in the growth of alfalfa under salt treatments and salt-free treatments. The results showed that, under salt stress and normal growth, the RE treatment had a positive effect on the plant damage and antioxidant systems and the negative effects of plant damage on alfalfa growth changed from −0.086 to −0.010 and −0.063 to 0.146, respectively. The positive effect of the antioxidant system on the growth of alfalfa increased from 0.031 to 0.112 and 0.016 to 0.107. However, under salt stress, REs changed the positive effects of the osmotic adjustment system, soil enzyme activity, and soil bacterial community on alfalfa growth from 0.132, 0.073, and 0.041 to negative effects of −0.043, −0.220, and −0.145, respectively.

Under normal growth, the RE treatment allowed the soil bacterial community to have positive and indirect effects on the growth of alfalfa by affecting the osmotic adjustment system (Figure 5b). On the contrary, the soil bacterial community without an RE treatment had indirect and negative effects on the growth of alfalfa by affecting the osmotic adjustment system (Figure 5a). Under salt stress, the RE treatment caused the soil bacterial community to have a positive and indirect effect on the growth of alfalfa by affecting soil enzyme activity (Figure 5d), while the non-RE treatment was the soil bacterial community without modification; its indirect and negative effects on the growth of alfalfa were affected by soil enzyme activity (Figure 5c).

## 3. Discussion

### 3.1. The Physiological Mechanism by Which REs Promote Alfalfa Growth

REs have been recognized for their potential to enhance plant growth, particularly under stressful conditions. In this study, when *Suaeda glauca* REs acted on alfalfa, compared with no treatment, its plant height increased by 47.70% to 76.57% and its fresh weight increased by 41.38% to 50.00% (Figure 1). In a study conducted by Kama, REs from maize (*Zea mays* L.) and soybean (*Glycine max*) were collected under typical intercropping conditions and subsequently applied to soybean plants subjected to irrigation with wastewater [47]. The observed outcomes demonstrated a significant increase in both the plant height and dry weight of soybean plants treated with the collected REs, aligning closely with the findings of this study. This increase may be attributed to the rise in osmotic adjustment substances and antioxidant enzyme activities in alfalfa, coupled with the decrease in its MDA activity. The elevation of osmotic adjustment substances can allow plants to grow in saline–alkaline soil, maintaining the water balance within cells [48]. Through the accumulation of specific organic and inorganic solutes, such as proline, mannitol, and ions, the osmotic concentration of the cell fluid increases to prevent excessive water loss [49]. Osmoregulation is closely related to cell growth and division, and excessive dehydration or cell expansion may interfere with these processes [50]. Antioxidant enzymes can scavenge the reactive oxygen species in plants, protecting plant cells [51]. Antioxidant enzymes are also involved in the synthesis and degradation of plant hormones such as gibberellin, indole acetic acid, and ethylene. Plant antioxidants can regulate key metabolic enzymes, influencing various cellular processes. This regulation influences the metabolic pathways, contributing to the overall homeostasis of plants [52], which ultimately affects plant growth and development. In Didi’s study, Nitraria tangutorum increased the activity of antioxidant enzymes, enhanced the accumulation of osmotic substances, reduced the content of MDA, and improved its own growth, aligning with the results of this study [53].

### 3.2. Effects of REs on the Physiology of Alfalfa

Under high-salt conditions, plants undergo ion imbalance and water deficiency, leading to osmotic stress, and osmotic regulation plays a crucial role in alleviating this pressure [54]. In this study, REs significantly increased the proline content in alfalfa under salt stress (Figure 2f), and its soluble sugar content was also significantly increased at 7 day and 14 day (Figure 2e). Xu et al. investigated the correlation between wheat REs and the growth of watermelon (*Citrullus lanatus*) [55]. Their findings indicated that wheat (*Triticum aestivuml*) REs positively influenced the levels of proline and soluble sugar in watermelon roots. This observation corresponds with the outcomes of the present study, reinforcing the notion that REs can improve the osmotic regulation of plants. REs may serve a dual purpose, regulating plants’ osmotic balance through distinct mechanisms. Firstly, they function as mediators in the intricate interactions between rhizosphere plants and microorganisms. By modulating this relationship, REs influence soil nutrient availability, microbial diversity, and metabolite production, ultimately facilitating plant osmotic regulation [56]. Additionally, REs have the potential to directly modulate the osmotic equilibrium within plant cells by regulating the deposition of solutes, such as amino acids [57]. This regulatory process effectively diminishes the osmotic potential of cells, thereby aiding in the maintenance of cellular hydration levels. The integration of these two mechanisms underscores the multifaceted role of REs in governing plant responses to osmotic stress within their surrounding soil environment. REs enhance the osmotic regulation of alfalfa and maintain cell functions, which is essential for alfalfa’s salt tolerance. The lipid oxidation reaction of the cell membrane can damage its structure and function, and MDA is the end product of lipid peroxidation. Therefore, its accumulation can be used as an index to measure the degree of stress and the extent of plant cell damage. In this study, the MDA content of the RE treatment was significantly lower than that of the CK (Figure 2a), while the activities of SOD, POD, and CAT in the RE treatment were higher than those in the CK (Figure 2b–d). Xiong isolated KLBMP4941, a strain elicited by *Limonium sinense* REs, and reintroduced it to *Limonium sinense*. Following this introduction, analyses revealed noteworthy increases in POD, SOD, and CAT activities within the treated plants [58]. The results reported in this paper corroborate those documented previously, indicating that REs have the capability to elevate the activity of antioxidant enzymes in plants. Consequently, this enhancement serves to mitigate the risk of damage to plant cell membranes. Zhu conducted a metabolomics analysis of cotton (*Gossypium hirsutum*) REs under varying levels of cadmium pollution. Their study revealed that the activities of SOD and CAT exhibited a negative correlation with the relative abundance of citric acid, green pigment, and catechin in the REs. Conversely, there was a positive correlation between the activities of SOD and CAT and the relative abundance of l-glutamic acid, l-asparagine, and 4-hydroxyphenylpyruvate [59]. Zhou et al. similarly observed a notable increase in the amino acid content within thyme (*Thymus mongolicus*) REs when the plants exposed to soil contaminated with cadmium and plumbum [60]. These findings imply that plant roots have the capacity to respond to abiotic stresses, such as cadmium pollution, by increasing their secretion of diverse amino acids. This response serves to bolster the antioxidant defense system within the plant, consequently enhancing its resilience against cadmium-induced stress.

### 3.3. Soil Microbial Mechanism of REs That Promotes Alfalfa Growth

REs are organic compounds that are the main nutrient sources of many soil microorganisms. Microorganisms are attracted to these organic compounds and accumulate in the rhizosphere area to form a rhizosphere microbial community [27]. In this study, REs did not significantly alter the diversity of the bacterial communities in alfalfa rhizosphere soil, but they significantly increased the abundance of beneficial bacterial species (Figure 4d). Zou’s investigation into the impact of citric acid, a key constituent of root exudates, on arsenic transformation and microbial communities across ten diverse paddy soils revealed a consistent pattern with the findings of this paper [61]. Despite no observed alteration in microbial diversity, there was a notable shift towards the prevalence of Clostridium-related bacteria, known for their role in regulating arsenic release. This change may be attributed to certain macromolecules in REs, such as mucus and polysaccharides, contributing to the formation of soil aggregates and maintaining the stability of microbial communities [62]. Moreover, REs have an attractive effect on beneficial bacteria or can provide them with a source of nutrition, making the soil more conducive to the growth of beneficial bacteria, and inducing the alfalfa rhizosphere to recruit more beneficial bacteria to resist the damage of salt stress. In previous studies, several beneficial microorganisms have been reported to improve plant growth and salt tolerance, including *Bacillus*, *Ochrobactrum*, and *Gemmatimonadota* [63]. For example, *Gemmatimonadota* can fix nitrogen in the atmosphere under hypoxia, increase soil nutrients, and improve soil structure, thereby improving the salt tolerance of plants [64]. It was found that *Gemmatimonadota_unclassified*, *Gemmata*, and *Gemmatimonas* gathered more in the 1.2% NaCl + RE treatment group than in the 1.2% NaCl treatment group (Figure 4d). Soil microorganisms produce most of the enzymes in soil, of which bacteria are an important source [65]. Soil bacteria produce soil enzymes and release them into the soil to help decompose organic matter and obtain nutrients [66]. Soil enzyme activity can reflect the direction and intensity of the biochemical reactions in soil and can directly or indirectly indicate the influence of REs on rhizosphere soil [67,68,69]. In this study, four important soil enzymes, S-UE, S-SC, S-CAT, and S-NP, were studied. They are closely related to soil nitrogen metabolism [70], organic matter metabolism [71], aerobic organism metabolism [72], and soil phosphorus availability [73]. The results showed that the activities of S-UE, S-SC, and S-CAT in rhizosphere soil decreased significantly after 1 day under salt stress (Figure 3a–c), which indicated that a high salt concentration adversely affected the metabolism of nitrogen, organic matter, and aerobic organisms in the rhizosphere soil of alfalfa. At 14 day, the activities of urease, invertase, and neutral phosphatase in rhizosphere soil increased significantly (Figure 3a,b,d), which may be the adaptive response of the plant and soil ecosystem to long-term salt stress. Shi et al. also reported this phenomenon [74]. The RE treatment further increased the activities of urease and neutral phosphatase in rhizosphere soil (Figure 3a,d), although not significantly. The high enzyme activity in rhizosphere soil is not only derived from the direct release of enzymes from plant roots, but also affected by soil microbial activity [75]. This is because REs contain a large number of compounds that act as signals for establishing and regulating plants’ interactions with microorganisms. The composition and activity of soil microorganisms further affect soil enzyme activity and plant functional characteristics [76,77,78]. Therefore, it has been suggested that REs provide additional carbon and energy to the microorganisms in saline soil, stimulate the growth and metabolic activities of microorganisms, and then increase the activities of urease and neutral phosphatase, which are helpful for plants to absorb nitrogen and phosphorus more effectively [76,78]. This phenomenon strongly corroborates the alleviation of salt stress in alfalfa, a change that is intricately linked to the modulation of soil microorganisms.

### 3.4. Effects of REs on the Bacterial Community in Alfalfa Rhizosphere Soil

Redundancy analysis (RDA) is a multivariate statistical method used to analyze the relationship between two sets of variables [79]. In this study, an RDA was carried out to reveal the environmental factors affecting the composition of the soil bacterial community in alfalfa (Figure 6a,b), with the abundance of the phylum and genus communities in alfalfa soil as the response variables and S-UE, S-SC, S-CAT, and S-NP as the explanatory variables. The RDA of the bacterial phylum level and soil enzyme activity (Figure 6a) showed that the cumulative explained variation of the first and second axes reached 45.07% and that the two axes could reflect the effect of soil enzyme activity on the soil’s bacterial community structure. The results of Figure 6b show that the cumulative explained variation of the first axis and the second axis reached 59.93%, reflecting the impact of soil enzyme activity on the soil bacterial community’s structure at the genus level. According to the length of the arrow connection, urease was the main environmental factor affecting the soil bacterial community’s structure. The R2 values of urease were 0.891 and 0.611, respectively, indicating that urease could explain 89.1% and 61.1% of the total variation of the soil bacterial phylum and genus communities. Therefore, urease may be the main factor affecting the soil bacterial community. In another study, the addition of urease inhibitors significantly increased the abundance of the 16S rRNA of bacteria and fungi, which proved that urease was the main factor affecting the soil bacterial community [80], which was the same as the results in this paper. Spearman correlation analysis is a statistical method used to measure the nonlinear relationship between two variables. For the exploratory analysis of many variable pairs, it is a better correlation measure because it is effective for nonlinear relationships and can eliminate the influence of other variables [81]. A Spearman correlation analysis was performed on the top 6 soil bacterial phyla and the top 10 bacterial genera and their environmental factors (Figure 6c,d). The results showed that soil enzyme activities affected the bacterial community’s composition and that soil urease and neutral phosphatase were significantly positively correlated with *Gemmatimonadota*, *Chloroflexi*, and *KD4-96_unclassified* (*p* < 0.01). *Gemmatimonadota* plays a role in the nitrogen cycle by fixing nitrogen and unlocking phosphorus and other trace elements in the soil, making them absorbable and utilizable by plants [82]. Urease also plays a vital role in the nitrogen cycle by catalyzing the hydrolysis of urea to ammonia and carbon dioxide [83]. The enhancement of urease activity increased the absorption of a key growth element, nitrogen, by alfalfa, and promoted the growth rate and productivity of alfalfa. In addition, it was also revealed that REs may promote the growth of alfalfa under salt stress by regulating the bacterial community structure in the soil, such as increasing the proportion of bacteria such as *Gemmatimonadota*, increasing the availability of nitrogen in the soil, and improving the resistance and resilience of the ecosystem by enhancing the resistance and growth ability of plants. Research has demonstrated that alterations in the soil bacterial community’s structure can impact soil health, agricultural productivity, and ecosystems [84]. Under salt stress, the core groups of bacterial communities often shift from eutrophication to oligotrophy, resulting in alterations in nutrient utilization patterns and other environmental factors [85]. *Gemmatimonadota* is an oligotrophic microorganism that can adapt to extreme environments such as low nutritional conditions and multiple stresses [86]. Studies have shown that *Gemmatimonadota* is a vector for a variety of resistance genes, including some heavy metal resistance genes, which can be transferred to other microorganisms that do not contain resistance genes [87]. The proliferation of bacteria has been identified as a crucial factor in augmenting soil resistance genes [88]. The increase and transfer of *Gemmatimonadota*’s resistance genes help to maintain the diversity of microbial communities in environmentally polluted soils and enhance the stability of ecosystems. The enhancement of microbial activity also helps to increase biodiversity and promote the function and service capacity of the entire ecosystem.

### 3.5. Comprehensive Analysis of the Effects of Plant Physiology and Soil Changes on the Growth of Alfalfa under RE Treatment

In this study, a partial least squares regression model (PLS-SEM) was used to study the effects of REs on plant damage, the osmotic adjustment system, antioxidant system, soil enzyme activity, and soil bacterial community interactions in the plant growth of alfalfa under salt stress (Figure 5). Reliability and validity assessments were conducted to effectively evaluate the PLS-SEM model’s reliability [89,90]. Since the data of this study meet the above conditions well, the factor loads of the main indicators in Table A1 all fulfill the requirements of structural validity [91]. The VIF was used to detect the multicollinearity of 20 variables. The results showed that the VIF values of all dominant variables were between 1 and 5, indicating that there was no serious covariance between elements [92]. The SRMR values of the CK, RE, 1.2% NaCl, and 1.2% NaCl + RE treatments were 0.088, 0.097, 0.095, and 0.092, respectively, indicating that the model had a good degree of fitting. In summary, the above indicators have basically reached their ideal value, which proves the rationality and reliability of the evaluation model. The models of the CK and RE treatments showed that the direct effects of the antioxidant enzyme system, plant damage system, and osmotic regulation system on the growth of alfalfa were transformed into strong positive effects after the RE treatment (Figure 5a,b). In the model of the 1.2% NaCl and 1.2% NaCl + RE treatments, REs had a direct positive effect on the growth of alfalfa by regulating the plant damage system and antioxidant system, but reduced the positive effect of the osmotic adjustment system on plant growth (Figure 5c,d). Under a salt stress treatment, alfalfa plays a role through its osmotic adjustment system and antioxidant enzyme system, with REs elevating the common role of antioxidant enzyme activity into a predominant role. The PLS-SEM analysis also revealed the complex interaction between REs and the alfalfa’s soil bacterial community, soil enzyme activity, and plant growth. In a normal growth environment, REs allowed the soil bacterial community to have a positive direct effect on soil enzyme activity, but soil enzyme activity had no obvious direct effect on plant growth (Figure 5a,b). It is possible that the effect of the improvement of enzyme activity on plant growth has not been fully demonstrated, and sometimes the improvement of soil conditions takes some time to be reflected in plant growth. In a salt stress environment, REs allowed the soil bacterial community to have an indirect positive effect on the growth of alfalfa through soil enzyme activity (Figure 5c,d). The importance of REs as a carrier between plants and the soil environment is due to their affect on the abundance and diversity of soil bacterial communities [93]. Soil bacterial communities will affect the activity of soil enzymes and ultimately affect the growth of plants [94]. In summary, plant–microbe interactions, mediated by REs, play a crucial role in enriching the soil’s structure and function, including soil nutrient acquisition [78].

## 4. Materials and Methods

### 4.1. Test Materials

*Medicago sativa* L.cv. ‘Dongnong No. 1’ and *Suaeda glauca* (Bunge) Bunge were used as test plants. Alfalfa was provided by Northeast Agricultural University, and *Suaeda glauca* was from local *Suaeda glauca* in Zhaodong City, Heilongjiang Province.

### 4.2. Experimental Design

#### 4.2.1. Collection of the REs of *Suaeda glauca*

After sterilization and disinfection, the seeds were placed in a constant-temperature incubator at 25 °C and cultured with vermiculite. The constant temperature was set at 25 °C, the relative humidity was 80%, the light intensity was 6000 lx, and the photoperiod was 12 h light/12 h dark. Hoagland solution was applied to the seeds. The pot experiment began 7 days post germination. After that, a certain number of 7-day-old seedlings of *Suaeda glauca* with the same size and growth vigor were selected and transplanted into a flowerpot containing tested saline soil (the flowerpot was 14 cm high, its lower diameter was 9.5 cm, and its upper diameter was 15 cm). The physical and chemical properties of *Suaeda glauca* soil were as follows: the pH was 9.01, organic matter content was 398.24 g kg^−1^, and total nitrogen content was 3.35 g kg^−1^. The ammonium nitrogen content was 45.55 mg kg^−1^, the nitrate nitrogen content was 29.48 mg kg^−1^, and the total phosphorus content was 0.213 g kg^−1^. The flowerpot was placed in a growth room with 60% humidity and a 12 h photoperiod, at 25 °C. The plants with consistent growth were sampled at 45 day. The complete roots of *Suaeda glauca* plants were collected using a sterile shovel and placed in sterile bags. The collection method of root exudates was according to the mixed improvement method, and the concentration was 15 plants per liter [95]. The stock solution of root exudates was stored in a refrigerator at −80 °C.

#### 4.2.2. Alfalfa Seedling Test

The sterilized alfalfa seeds were planted in plastic trays (41 cm long, 41 cm wide, and 5 cm high). After 21 days of seed germination, 10 seedlings were planted in each pot of 700 g of soil (flowerpot height of 14 cm, lower diameter of 9.5 cm, upper diameter of 15 cm), and alfalfa seedlings were regularly watered. The physical and chemical properties of the alfalfa soil were as follows: the pH was 6.51, organic matter content was 203.68 g kg^−1^, total nitrogen content was 2.04 g kg^−1^, ammonium nitrogen content was 22.22 mg kg^−1^, nitrate nitrogen content was 60.89 mg kg^−1^, total phosphorus content was 1.18 g kg^−1^. After 20 days, the pots were randomly divided into 6 groups: the CK (0% NaCl), RE (*Suaeda glauca* root exudates), 0.4% NaCl (salt concentration of 4 g/kg), 0.4% NaCl + RE, 1.2% NaCl (salt concentration of 12 g/kg), and 1.2% NaCl + RE treatments. The salt stress treatment group was evenly irrigated with a NaCl solution. In order to avoid the acute stress of high salt in plants, the low salt concentration group reached its salt stress concentration on the 1st day, and the high salt concentration group reached its salt stress concentration on the 3rd day (while the high salt concentration group was irrigated with NaCl solution for 3 days, the low salt concentration group was irrigated with NaCl solution to reach its salt stress concentration at the same time). The other treatment groups and the salt stress treatment group were irrigated with the same volume of ultrapure water, and the lower parts of the flowerpots were in trays. The solution in the tray was collected and returned to the soil every 2 days. After 24 h, 100 mL of REs was applied to the RE treatment group and 100 mL of ultrapure water was applied to the CK treatment group, which formed the 0th day of the experiment. The pots were placed in the growth room. For the environmental temperature, humidity, and light conditions of the test, refer to Section 4.2.1. At 1 day, 7 day, and 14 day, three flowerpots were randomly selected from each group for sampling and placed in a refrigerator at −80 °C for testing, and their growth and physiological indexes were measured.

#### 4.2.3. Alfalfa Rhizosphere Soil Test

In this study, 1 day and 14 day alfalfa rhizosphere soils were selected to explore the difference between the soil conditions before planting and the soil conditions after the end of the plant growth period. In the rhizosphere soil experiment, the rhizosphere soil of the CK, RE, 1.2% NaCl, and 1.2% NaCl + RE treatment groups was used explored. The rhizosphere soil of alfalfa was sampled with a sterile shovel and placed in an −80 °C refrigerator for soil enzyme activity and soil bacterial diversity determination.

### 4.3. Determination of the Alfalfa Growth Index

Determination of aboveground plant height: the natural height of alfalfa growth in the pot was measured using a tape measure, and 6 seedlings with consistent growth were selected from each pot. Determination of aboveground fresh weight: The aboveground part of the plants was cut open with sterile scissors, and the fresh weight of the aboveground part was measured after the balance was adjusted to zero. Six seedlings with consistent growth were selected from each pot.

### 4.4. Determination of the Physiological Indexes of Alfalfa

#### 4.4.1. MDA Content

The determination of MDA in alfalfa leaves was performed using a 96-well microassay kit produced by Suzhou Keming Biotechnology Co., Ltd. (Suzhou, China), using 0.1 g of leaf powder ground in liquid nitrogen and adding reagents according to the kit instructions (www.cominbio.com, accessed on 10 March 2023). The difference in absorbance at 532 nm and 600 nm was calculated using a K6600-a (Beijing Kaiao Technology Development Co., Ltd., Beijing, China) microplate reader, and the calculation formula was based on the kit instructions.

#### 4.4.2. SOD, POD, and CAT Activity

The activities of SOD (EC 1.15.1.1), POD (EC 1.11.1.7), and CAT (EC 1.11.1.6) in alfalfa leaves were determined using a 96-well microassay kit produced by Suzhou Keming Biotechnology Co., Ltd. (Suzhou, China), using 0.1 g of leaf powder ground in liquid nitrogen was used and adding reagents according to the kit instructions (www.cominbio.com, accessed on 10 March 2023). The K6600-a microplate reader (Beijing Kaiao Technology Development Co., Ltd., Beijing, China) was used to determine the absorbance of the corresponding enzyme’s activity at 560 nm, 470 nm, and 405 nm, respectively. The calculation formula was based on the kit instructions.

#### 4.4.3. Soluble Sugar Content

The determination of the soluble sugars in alfalfa leaves was performed using a 96-well micro-method kit produced by Suzhou Keming Biotechnology Co., Ltd. (Suzhou, China), using 0.1 g of leaf powder ground in liquid nitrogen and adding reagents according to the kit instructions (www.cominbio.com, accessed on 10 March 2023). The K6600-a (Beijing Kaiao Technology Development Co., Ltd., Beijing, China) microplate reader was used to compare the color at a wavelength of 620 nm, and the absorbance value was recorded. The calculation formula was based on the kit instructions.

#### 4.4.4. Free Proline Content

The determination of the free proline in alfalfa leaves was determined using a 96-well micromethod kit produced by Suzhou Keming Biotechnology Co., Ltd. (Suzhou, China) and 0.1 g of leaf powder ground in liquid nitrogen, according to the instructions for use (www.cominbio.com, accessed on 10 March 2023). We added the reagent and used the K6600-a (Beijing Kaiao Technology Development Co., Ltd., Beijing, China) microplate reader to compare the color at a wavelength of 520 nm, record the absorbance value, and calculate the formula with reference to the kit instructions.

### 4.5. Determination of Alfalfa Rhizosphere Soil

#### 4.5.1. Soil Enzyme Activity

The activities of S-UE, S-SC, S-NP, and S-CAT in the rhizosphere soil of alfalfa under salt stress were determined using *Suaeda glauca* root exudates. The 96-well micro-assay kit produced by Suzhou Keming Biotechnology Co., Ltd. (Suzhou, China) was used to weigh 0.1 g of fresh soil samples passed through a 100 mesh sieve. The reagent was added according to the kit instructions (www.cominbio.com, accessed on 10 March 2023), and the absorbance values of the corresponding enzyme activities were measured at 578 nm, 510 nm, 660 nm, and 240 nm using K6600-a (Beijing Kaiao Technology Development Co., Ltd., Beijing, China) microplate reader. The calculation formula was determined by referring to the kit instructions.

#### 4.5.2. Soil DNA Extraction, PCR Amplification, and 16S rDNA Sequencing Data Processing

According to the manufacturer’s instructions, the DNA of different samples was extracted by the CATB method, and then the quality of the DNA extraction was detected by agarose gel electrophoresis and the DNA was quantified using an ultraviolet spectrophotometer. Primer 341F (5′-CCTACGGGNGGCWGCAG-3′) was used. 805R (5′-GACTACHVGGGTATCTAATCC-3′) was used for the PCR amplification of bacterial 16S rDNA hypervariable regions V3-V4. PCR products were purified using AMPure XT beads (Beckman Coulter Genomics, Danvers, MA, USA) and quantified by Qubit (Invitrogen, Waltham, MA, USA). The PCR products were detected by 2% agarose gel electrophoresis and recovered using an AMPure XT beads recovery kit. The Illumina NovaSeq PE250 platform was used for sequencing at the Hangzhou Lianchuan Biotechnology Co., Ltd., Hangzhou, China.

### 4.6. Partial Least Squares Regression Model (PLS-SEM)

PLS-SEM is used to estimate the causal network between latent variables. Latent variables can be represented by a set of explicit variables. An observable variable is a quantity that can be observed or measured directly. The structural model (or internal model) is a path diagram that reflects the relationship between the effects of latent variables [96,97]. There are many factors affecting the growth of alfalfa under salt stress, including plant damage, plant osmotic regulation, and antioxidant mechanisms, as well as soil nutrients and microbial conditions. These factors vary with different treatment methods. In this study, plant height, plant fresh weight, MDA content, proline content, soluble sugar content, SOD activity, POD activity, CAT activity, S-UE, S-SC, S-CAT, S-NP, and the soil bacterial community (PCA analysis of the main soil bacterial genera and the extraction of their main components) were used as significant variables. Plant growth, plant damage, the osmotic adjustment system, antioxidant system, soil enzyme activity, and soil bacterial community were used as potential variables in the PLS-SEM model.

Soil bacteria not only participate in nutrient cycling, such as the nitrogen cycle and phosphorus cycle, affecting nutrient availability and thus affecting soil enzyme activity [98], but also enhance the plant uptake of water and mineral nutrients, helping to maintain plants’ osmotic adjustment [99]. Soil bacteria (such as rhizobium PGPR, which promotes plant growth) can also enhance plants’ osmotic adjustment and antioxidant capacity by up-regulating antioxidant defense pathways and increasing the accumulation of osmolytes [100]. Soil enzymes affect the osmotic regulation and antioxidant system of plants through their effects on the soil nutrient supply, organic matter decomposition, and soil pH [101]. The antioxidant system and osmotic adjustment system of plants can reduce plant damage [99]. The PLS-SEM model was created based on the above conclusions.

SmartPLS (SmartPLS: Ringle, C. M., Wende, S., and Becker, J.-M. 2022. ‘SmartPLS 4.’ Oststeinbek: SmartPLS GmbH, http://www.smartpls.com, accessed on 14 March 2023) was used for PLS-SEM analysis. The PLS-SEM algorithm adopted standardized data, a weighted scheme adopted path, and the initial weight was set as default.

### 4.7. Data Analysis

A one-way analysis of variance was performed using SPSS26.0 for the comparison between groups. Duncan’s method was used for a significant difference test (*p* < 0.05), and GraphPad Prism 9.0 was used for mapping. Bioinformatics analysis was performed using the OmicStudio tool (https://www.omicstudio.cn, accessed on 17 March 2023). A Venn diagram was drawn based on R version 4.1.3 on the OmicStudio platform. It was used to display the number of OTUs shared between samples and intuitively explain the overlap of OTUs between each sample. The top 30 phyla and genera with higher abundances were screened from the OTUs of bacteria, and the abundance composition histograms and heat maps of the top 30 phyla and genera under different treatments were drawn based on R version 3.6.3 on the OmicStudio platform. Spearman correlation analysis and RDA were performed using the OmicStudio platform R version 3.6.3 to evaluate the correlation between rhizosphere soil enzyme activity and the top 6 bacterial phyla and the top 10 bacterial genera. Based on R version 3.6.3, a correlation clustering heat map and redundancy analysis map were drawn (*p* < 0.05 indicated significant correlation, *p* < 0.01 indicated extremely significant correlation). The composition of soil bacterial communities in the PLS-SEM model was obtained through the principal component analysis of the top 100 bacterial genera, performed using SPSS26.0. The main components were extracted based on the principle that the cumulative variance contribution rate was not less than 80%, and the regression method was used to calculate the score coefficients of each factor.

## 5. Conclusions

The findings of this study provide empirical support for the hypothesis that REs from stress-tolerant plants can indeed facilitate the growth of other plants under stressful conditions. Indeed, the observed enhancement in the growth of alfalfa under salt stress can be attributed to multiple factors that were enhanced by the REs of *Suaeda glauca.* These factors include the augmentation of soil enzyme activity, which aids in nutrient availability, and the positive modulation of the microbial community’s composition. Additionally, the REs contribute to the improvement of alfalfa’s osmotic regulation and antioxidant defenses, enhancing its resilience to salt stress. The RDA conducted in this study indicated that urease is the primary factor influencing the composition of the soil bacterial community. Moreover, the Spearman correlation analysis unveiled significant relationships between soil enzyme activity and the composition of bacterial communities. Notably, S-UE and S-NP exhibited significant positive correlations with taxa such as *Gemmatimonadota*, *Chloroflexi*, and *KD4-96_unclassified*. The PLS-SEM model indicated that, when subjected to salt stress, alfalfa primarily relies on its osmotic regulation system and antioxidant enzyme system to cope with the stress. However, the introduction of REs alters this dynamic, shifting the predominant role in mitigating the effects of salt stress on alfalfa to antioxidant enzyme activity. These findings offer valuable insights that can aid in the development of sustainable strategies aimed at bolstering the resilience of important crops. Given the challenges posed by global climate change, such strategies are crucial for ensuring food security and agricultural sustainability in the face of increasingly adverse environmental conditions.

## Figures and Tables

**Figure 1 plants-13-00752-f001:**
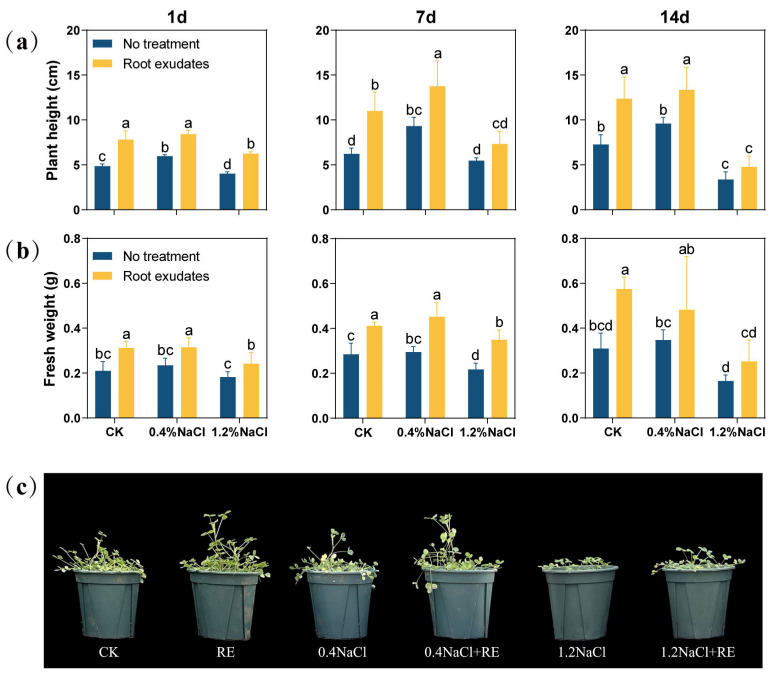
Effects of *Suaeda glauca* root exudates (REs) on the growth of alfalfa at 1 day, 7 day, and 14 day of salt stress. (**a**) Shoot height; (**b**) aboveground fresh weight; (**c**) plant growth phenotype chart at 14 d. CK represents a 0% NaCl concentration. No treatment represents the application of ultrapure water equivalent to REs, and root exudates represents the application of REs. Different lowercase letters represent significant differences between different treatments at the same time (*p* < 0.05).

**Figure 2 plants-13-00752-f002:**
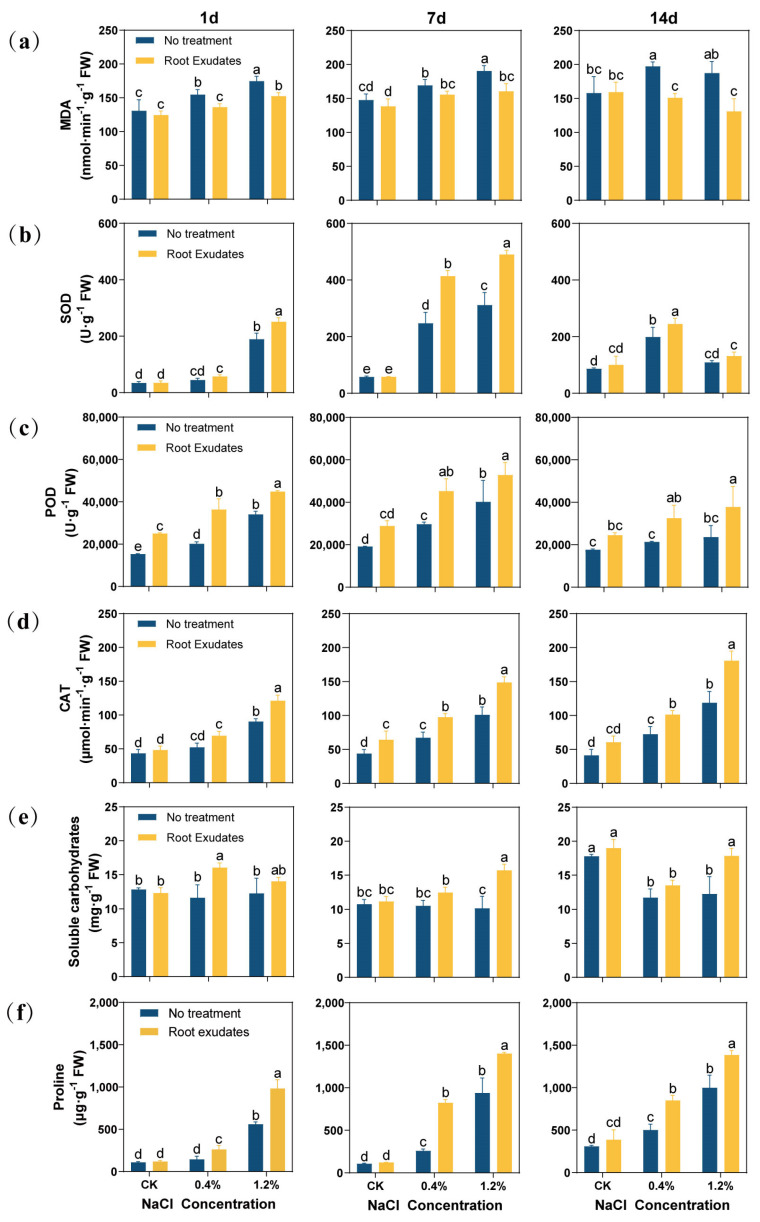
Effects of *Suaeda glauca* root exudates on membrane peroxidation, antioxidant enzymes, and the osmotic substances in alfalfa leaves at 1 day, 7 day, and 14 day of salt stress. (**a**) Malondialdehyde content (MDA); (**b**) superoxide dismutase (SOD); (**c**) peroxidase (POD); (**d**) catalase (CAT); (**e**) soluble sugar content; (**f**) proline content. CK represents a 0% NaCl concentration. No Treatment represents the application of ultrapure water equivalent to the REs, and Root Exudates represents the application of REs. Different lowercase letters represent significant differences between different treatments at the same time (*p* < 0.05).

**Figure 3 plants-13-00752-f003:**
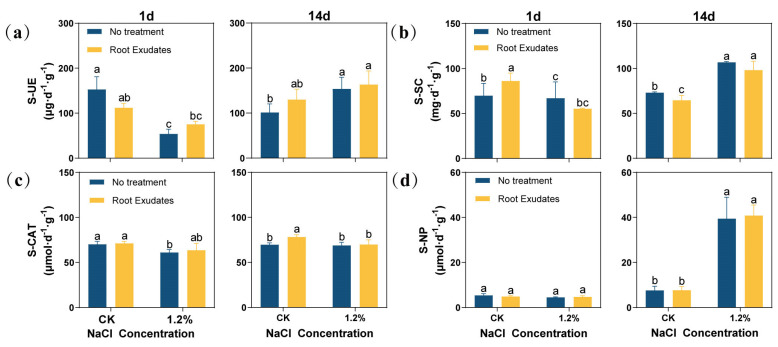
Effects of *Suaeda glauca* root exudates on the activities of urease, sucrase, catalase, and neutral phosphatase in the rhizosphere soil of alfalfa at 1 day, 7 day, and 14 day of salt stress. (**a**) Soil urease (S-UE) activity; (**b**) soil sucrase (S-SC) activity; (**c**) soil catalase (S-CAT) activity; (**d**) soil neutral phosphatase (S-NP) activity. CK represents a 0% NaCl concentration. No Treatment represents the application of ultrapure water equivalent to the REs, and Root Exudates represents the application of REs. Different lowercase letters represent significant differences between different treatments at the same time (*p* < 0.05).

**Figure 4 plants-13-00752-f004:**
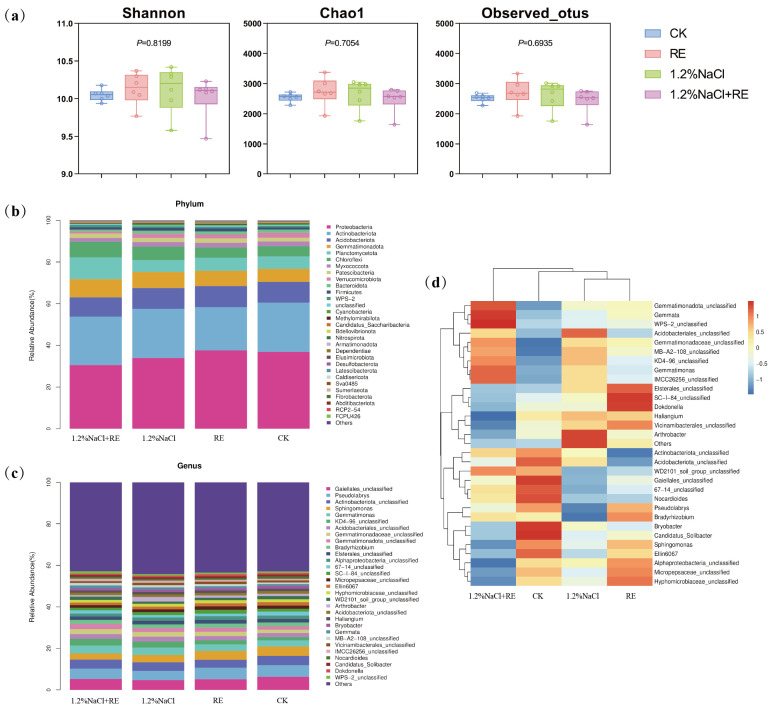
Effects of *Suaeda glauca* root exudates on the soil microbial community structure of alfalfa on the 14th day of salt stress. (**a**) The soil bacterial diversity index of different treatments; (**b**) the relative abundance of the top 30 most abundant bacterial phyla in the bacterial community structure of different treatments; (**c**) the relative abundance of the top 30 most abundant bacterial genera in the bacterial community structure of different treatments; (**d**) the distribution of the soil bacterial community (genus-level) in different treatments. CK and 1.2% NaCl indicated the application of ultrapure water equivalent to the REs, and RE and 1.2% NaCl + RE indicated the application of REs.

**Figure 5 plants-13-00752-f005:**
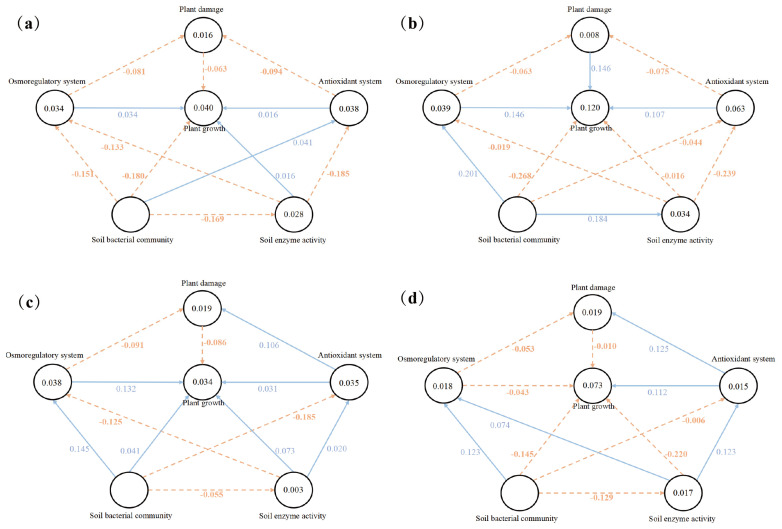
The PLS-SEM model diagram shows the effects of the CK (**a**), RE (**b**), 1.2% NaCl (**c**), and 1.2% NaCl + RE (**d**) treatments’ latent variables (plant damage, osmotic adjustment system, antioxidant system, soil enzyme activity, and soil bacterial community interactions) on plant growth. The arrows indicate the connections between latent variables. The orange dotted lines indicate a negative correlation, and the blue solid lines indicate a positive correlation.

**Figure 6 plants-13-00752-f006:**
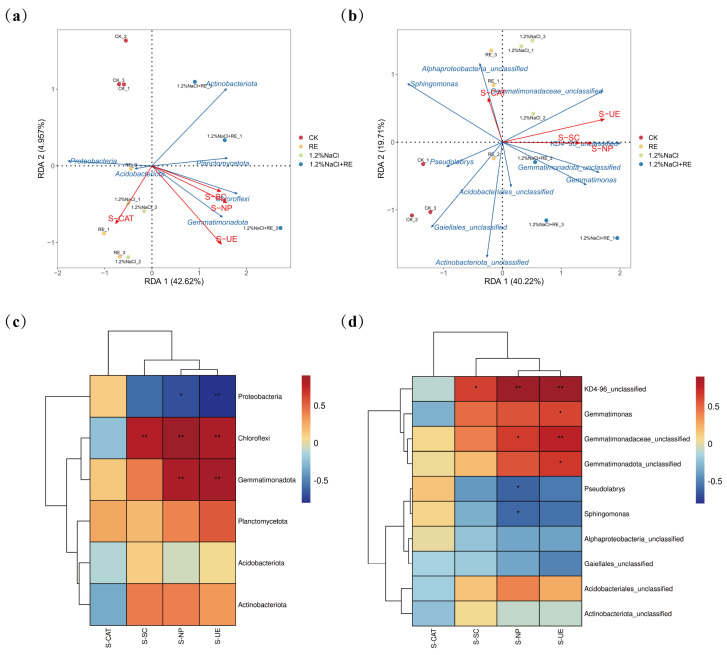
Relationship between the soil bacterial community’s structure and environmental factors of *Suaeda glauca* root exudates on the 14th day of alfalfa growing under salt stress. (**a**) Redundancy analysis (RDA) of the rhizosphere soil’s bacterial phyla (6 most abundant phyla) and soil enzyme activity in a single sample of different treatments; (**b**) redundancy analysis (RDA) of the rhizosphere soil’s bacterial genera (10 most abundant genera) and soil enzyme activities in a single sample of different treatments; (**c**) Spearman correlation between soil bacterial community (6 most abundant phyla) and soil enzyme activity; (**d**) Spearman correlation between soil bacterial community (10 most abundant genera) and soil enzyme activity. The blue arrows represent the dominant bacteria of the soil, the red arrows represent the environmental factors, and the length of the arrows represents the influence of the environmental factors on the dominant bacteria. The longer the length of the arrow, the greater the influence. CK and 1.2% NaCl were plied with the same amount of ultrapure water as REs, and the RE and 1.2% NaCl + RE treatments were plied with REs. S-CAT, S-SC, S-NP, and S-UE represent soil catalase, soil invertase, soil neutral phosphatase, and soil urease, respectively. * 0.01 ≤ *p* < 0.05, ** 0.001 ≤ *p* < 0.01.

## Data Availability

The data presented in this study are available on request from the corresponding author. The data are not publicly available due to privacy.

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
