# Peer review of "The Multiple Promoting Effects of Suaeda glauca Root Exudates on the Growth of Alfalfa under NaCl Stress"

_plants, 2024, doi:10.3390/plants13060752_

Round 1

Reviewer 1 Report

Comments and Suggestions for Authors

The article in question is very interesting and constitutes an interesting attempt to solve the problem of salinity in agricultural soil. The manuscript is written in a logical and orderly manner. However, it lacks clear information about why the authors chose these specific sodium chloride concentrations (salinity levels) for their research. How do these concentrations relate to naturally occurring soils?

Reviewer 2 Report

Comments and Suggestions for Authors

Dear Authors,

Thank you for this manuscript, which significantly contributes to our understanding of the promoting effects of Suaeda glauca root exudates on alfalfa plant growth under salinity conditions. The findings of this study demonstrate that these root exudates play a crucial role in enhancing the tolerance and biomass of alfalfa when subjected to salt stress. The mechanism of action underlying these effects appears to involve an increase in osmotic compound content and the activity of antioxidant enzymes, leading to a reduction in malondialdehyde levels in alfalfa. This, in turn, helps with the integrity and functionality of cell membranes while enhancing the plant's ability to scavenge reactive oxygen species. Furthermore, your results suggest that Suaeda glauca root exudates positively impact the abundance of beneficial bacteria and urease activity in the soil, indirectly facilitating nitrogen uptake by alfalfa. While the manuscript is well-written, I recommend several modifications before the publication: Introduction Section: Ensure that aspects discussed in the introduction are elaborated upon in the discussion section. This includes detailing the significance of root exudates in osmoregulation, antioxidant activity, and soil microbial diversity in the introduction section. Strengthening these connections will bolster the hypothesis. Scientific Names Formatting: Verify that all scientific names are italicized throughout the manuscript for consistency and adherence to formatting standards. Figure Legends: Provide comprehensive descriptions in the legends of all figures, including a detailed account of treatments. For example, specify the treatment represented by "CK" in Figure 1, and in Figure 3: describe the enzymes with abbreviations used. These adjustments will enhance the clarity and completeness of the manuscript for their publication.

Reviewer 3 Report

Comments and Suggestions for Authors

Authors of the manuscript “Multiple promoting effects of Suaeda glauca root exudates on the growth of alfalfa under NaCl stress” observed that urease affected the composition of soil bacterial community. Partial least squares structural equation model (PLS-SEM) revealed that RE had a direct positive effect on alfalfa growth under salt stress by regulating plant injury and antioxidant system, and soil bacterial community had an indirect positive effect on alfalfa growth through soil enzyme activity.

Manuscript is well written. However, authors need to address the following issues:

1.      L11: Please revise the sentence “However, there are few studies on the effects of root exudates on the stress resistance of another plant”.

2.      L12: Please revise the sentence “The effects of root exudates (RE) of Suaeda glauca on the growth of alfalfa seedlings under salt stress were studied”.

3.      L13: Please revise the sentence “The results showed that the aboveground height and fresh weight of alfalfa increased by 47.72% and 53.39% respectively after 7 days of 0.4%NaCl treatment”.

4.      L17: Don’t start a sentence with an abbreviation or digit.

5.      Arrange keywords alphabetically. Keywords must be different from title to enhance search ability and findability. Select words that describes the highlight/novelty of the research.

6.      L37; 65….: Scientific names are always italicized. Moreover, write the complete name of an organism or term before writing its abbreviation (please check name of enzymes and biological agent etc.). Afterwards, no need to write the complete term in a section. Maintain uniformity.

7.      Briefly discuss the current state of global agriculture and the increasing threat of salinity due to climate change. This will help readers understand the urgency and significance of the issue.

8.      L439, 452: Please revise the sentences.

9.      There are many language/ grammatical mistakes in the manuscript.

10.  The study did not mention specific details about the environmental conditions in which the experiments were conducted, such as temperature, light, and humidity. Variability in environmental conditions could affect the results and interpretation of the findings, and therefore should be carefully considered and reported.

11.  Discussion section could benefit from more elaboration on the implications of changes in crop root-associated bacterial communities under salt stress. Discuss how these changes affect plant health, nutrient cycling, and overall ecosystem functioning.

Comments on the Quality of English Language

There are grammatical/ language mistakes.

Round 2

Reviewer 3 Report

Comments and Suggestions for Authors

Dear Authors,

Please check appropriate use and placement of abbreviations in abstract and other sections. Discussion section still needs improvement. Please add some logics for your findings.

Comments on the Quality of English Language

Minor language/ grammatical improvement is required.
